cellular biology, developmental biology, physiology

developmental hypoxia, environmental hypoxia, developmental programming, cardiomyocyte, ectotherm, phenotypic plasticity

**Author for correspondence:**
Ilan M. Ruhr
e-mail: ilan.ruhr@manchester.ac.uk

# Developmental plasticity of cardiac anoxia-tolerance in juvenile common snapping turtles (*Chelydra serpentina*)

Ilan M. Ruhr[1], Heather McCourty[1], Afaf Bajjig[1], Dane A. Crossley II[2], Holly A. Shiels[1] and Gina L. J. Galli[1]

[1]Division of Cardiovascular Sciences, School of Medical Sciences, University of Manchester, Manchester M13 9NT, UK
[2]Department of Biological Sciences, University of North Texas, Denton, TX 76203, USA

(iD) IMR, 0000-0001-9243-7055

For some species of ectothermic vertebrates, early exposure to hypoxia during embryonic development improves hypoxia-tolerance later in life. However, the cellular mechanisms underlying this phenomenon are largely unknown. Given that hypoxic survival is critically dependent on the maintenance of cardiac function, we tested the hypothesis that developmental hypoxia alters cardiomyocyte physiology in a manner that protects the heart from hypoxic stress. To test this hypothesis, we studied the common snapping turtle, which routinely experiences chronic developmental hypoxia and exploits hypoxic environments in adulthood. We isolated cardiomyocytes from juvenile turtles that embryonically developed in either normoxia (21% $O_2$) or hypoxia (10% $O_2$), and subjected them to simulated anoxia and reoxygenation, while simultaneously measuring intracellular $Ca^{2+}$, pH and reactive oxygen species (ROS) production. Our results suggest developmental hypoxia improves cardiomyocyte anoxia-tolerance of juvenile turtles, which is supported by enhanced myofilament $Ca^{2+}$-sensitivity and a superior ability to suppress ROS production. Maintenance of low ROS levels during anoxia might limit oxidative damage and a greater sensitivity to $Ca^{2+}$ could provide a mechanism to maintain contractile force. Our study suggests developmental hypoxia has long-lasting effects on turtle cardiomyocyte function, which might prime their physiology for exploiting hypoxic environments.

## 1. Introduction

Developmental plasticity refers to the process whereby genetically similar individuals display substantially different phenotypes, depending on the environmental conditions that they experience during early life [1]. Often, the mechanism underlying developmental plasticity involves epigenetic modifications which not only persist into adulthood that might be inherited by subsequent generations [2]. Consequently, fluctuations in the developmental environment can have enduring and far-reaching effects on animal populations.

Developmental plasticity has ecological relevance for ectothermic vertebrates that routinely experience environmental fluctuations during early life. In particular, oviparous (egg-producing) reptiles lay their eggs in subterranean nests that can become progressively hypoxic, as a result of combined changes in environmental gas conductance, rising egg-mass metabolism and metabolic activity of microorganisms [3,4]. The extent of hypoxia is dependent on nest shape and egg location (e.g. eggs closer to the surface will be less likely to become hypoxic), and some species might be subjected to tensions as little as 11% $O_2$ [3]. Such low levels of developmental $O_2$ are known to produce pathological phenotypes in mammals and birds [5,6]. By contrast, chronically

hypoxic alligator and turtle embryos are more tolerant of a subsequent acute hypoxic stress [7] and are able to withstand lower levels of hypoxia, without altering metabolism (lower critical $O_2$ tension), compared with control embryos [8]. These studies suggest early exposure to developmental hypoxia can programme hypoxia-tolerant phenotypes in reptiles. This phenomenon would be particularly beneficial for species that actively exploit hypoxic environments in adulthood, such as freshwater turtles [9,10]. For these organisms, developmental hypoxia might act as an important environmental cue that primes their physiology for a future life in hypoxic environments.

Hypoxic survival is critically dependent on the maintenance of cardiac function for the delivery of nutrients and the removal of waste. Therefore, developmental programming of the heart might provide a mechanism to enhance turtle hypoxia-tolerance. Numerous studies have shown developmental hypoxia alters aspects of embryonic and juvenile turtle cardiac structure and function, including intrinsic heart-rate and the expression of receptors involved in cardiac regulation [11–15]. Even if offspring that developed in hypoxia are maintained postnatally in normoxic conditions, differences in turtle ventricular structure and function have been observed in adulthood [14,15]. Considering that some of these studies were conducted in isolated heart preparations, the collective literature suggests developmental hypoxia alters the intrinsic properties of turtle cardiomyocytes.

In this study, we hypothesized that developmental hypoxia primes turtle cardiomyocyte physiology in a manner that protects the cell from hypoxic injury. To test this hypothesis, we assessed cardiomyocyte anoxia-tolerance in juvenile turtles that were previously exposed to developmental hypoxia (10% $O_2$) or normoxia (21% $O_2$). To gain mechanistic insight, we investigated intracellular $Ca^{2+}$, pH and reactive oxygen species (ROS) homeostasis, which are known to be associated with the hypoxic injury. In mammalian cardiomyocytes, previous studies have shown anoxia causes a profound intracellular acidosis and a reduction in ATP production, which eventually leads to intracellular $Ca^{2+}$ overload [16]. Even if the cell survives the anoxic stress, reoxygenation causes a burst of ROS that triggers apoptotic and necrotic cell death [16]. Therefore, we hypothesized that cardiomyocytes from turtles previously exposed to developmental hypoxia would possess adaptations in pathways associated with $Ca^{2+}$, pH and ROS management.

# 2. Material and methods

## (a) Animals

Common snapping turtle (Chelydra serpentina) eggs were collected from the wild in Minnesota, USA, and transported to the University of North Texas for incubation. Two eggs from individual clutches were staged to determine the age. At approximately 20% development, eggs were randomly assigned to either atmospheric $O_2$ (21% $O_2$) or hypoxic (10% $O_2$) cohorts. We acknowledge that there is no 'normal' $O_2$ tension for developing turtles; however, for the purposes of this manuscript, 21% and 10% $O_2$ groups will be designated 'normoxic, N21' and 'hypoxic, H10', respectively. Incubations lasted no more than 55 days and all eggs were maintained at a female-determining 30°C, since sex determination during embryonic development is temperature-dependent in turtles [13,17,18]. Upon hatching,

all turtles were housed in common, normoxic (21% $O_2$) conditions at 26°C. After seven months, the hatchlings were transported by air freight to the University of Manchester, UK. Hatchlings were individually housed in normoxic conditions, at room temperature, until experimentation, when aged between 15 and 24 months.

## (b) Solutions

The compositions of all solutions in this study are given in electronic supplementary material, table S1. To our knowledge, plasma ion concentrations of snapping turtles at room temperature in normoxia or anoxia have not been determined experimentally. Therefore, we designed our saline solutions according to plasma ion concentrations measured in red-eared slider turtles (Trachemys scripta) breathing 21% $O_2$, at 25°C [19]. To reveal differences in anoxia-tolerance between the two phenotypes [section (e)], we subjected the cardiomyocytes to an anoxic saline that combined the three main elements of in vivo anoxia: zero $O_2$, $CO_2$ retention and lactic-acid build-up. The precise values were based on gas and plasma ion concentrations measured in T. scripta, after 4 h of anoxia, at 25°C [19]. While we acknowledge that other plasma ions change with in vivo anoxia (e.g. $Ca^{2+}$, $Mg^{2+}$ and $K^+$), our anoxic challenge allowed us to reveal differences between the phenotypes, by changing a limited set of variables. All chemicals were purchased from Sigma-Aldrich, unless otherwise stated.

## (c) Cardiomyocyte isolation

Ventricular cardiomyocytes were isolated by enzymatic dissociation, as previously described [20]. Briefly, the heart was initially perfused with isolation solution at 28–30°C for 8–10 min (to remove blood and debris) and then with enzymatic dissociation solution for 26–30 min. After perfusion, the ventricle was minced and individual ventricular cardiomyocytes were released by gentle agitation; cardiomyocytes were suspended in isolation solution and maintained at room temperature (22°C), for up to 8 h.

## (d) Morphometric analysis

Cardiomyocyte morphometrics were determined with confocal microscopy. Freshly isolated cells were incubated with 1% wheat germ agglutinin, in PBS, for 60 min, at room temperature. Consecutive plane scans ($x$–$y$) were made through the cell to make a three-dimensional model ($z$-stack), from which cell length, width, depth and volume were calculated, using ZEN imaging software (Zeiss Microscopy GmbH, Jena, Germany).

## (e) Simulated anoxia and reoxygenation

The experimental set-up was designed to measure ventricular cardiomyocyte function with epifluorescent microscopy, during three distinct experimental periods: normoxia, an anoxic challenge and reoxygenation (figure 1a). Cells were placed in a perfusion bath (model RC-21BRFS, Warner Instrument, Hamden, CT; figure 1c) that was equipped with electrodes, for field stimulation, which were connected to a stimulator (model SD-9, Grass Instruments, Astro-Med, West Warwick, RI, USA). The bath was placed in a custom-built environmental chamber (Bold-Line Top Stage Incubator, model H301-NIKON-TI-SR, Okolab, Ottaviano, NA, Italy; figure 1d) that was fitted with an $O_2$ probe (mini-sensor model TROXF1100 and FireStingO2 optical $O_2$ meter, PyroScience GmbH, Aachen, Germany). The chamber was connected to external gas cylinders and perfusion lines with gas-tight tubing. For epifluorescent imaging, the chamber was mounted to an inverted microscope (model Eclipse TE-2000U, Nikon, Surrey, UK) that was coupled to an Optoscan

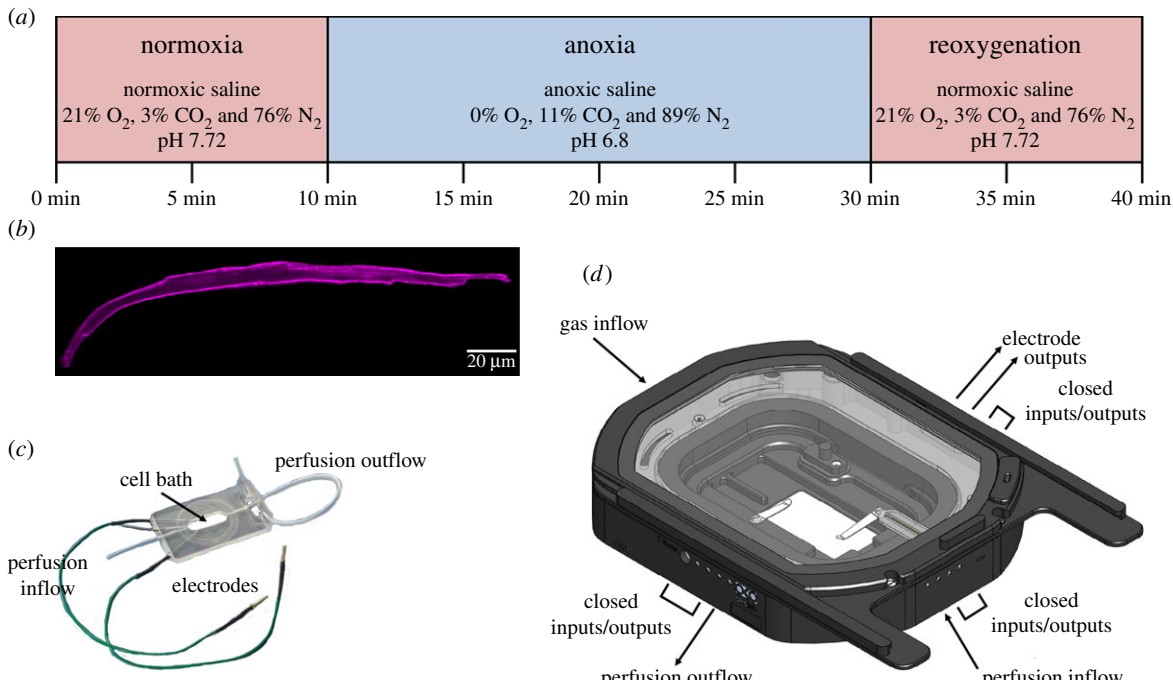

**Figure 1.** Experimental protocol and equipment for subjecting turtle cardiomyocytes to an anoxic challenge. (*a*) Ventricular cardiomyocytes from juvenile turtles that embryonically developed in normoxia (21% $O_2$) or chronic hypoxia (10% $O_2$) were subjected to a 40-min experimental protocol to determine the effects of simulated normoxia, anoxia and reoxygenation (same treatment as normoxia) on cell shortening, $[Ca^{2+}]_i$, $pH_i$ and ROS production. (*b*) Isolated cardiomyocytes (image collected from a confocal microscope) were placed onto (*c*) a closed-cell bath, within (*d*) a custom-built chamber, in which the cardiomyocytes were subjected to the two treatments. Images courtesy of Warner Instruments (model RC-21BRFS) and Okolab (custom-built bold-line top stage incubator). (Online version in colour.)

photomultiplier tube, monochromator and high-intensity xenon arc lamp (Cairn Research Instruments, Faversham, UK). Signals were digitized with a Digidata 1440A and analysed with pCLAMP 10 software (Axon Instruments, Sunnyvale, CA).

## (f) Experimental protocol

Cardiomyocytes were loaded with the AM-ester, cell-permeant fluorescent indicators (Invitrogen, Loughborough, UK) Fura-2 AM, BCECF and dihydroethidium (DHE) to measure $[Ca^{2+}]_i$, $pH_i$ and ROS, respectively, at room temperature. $[Ca^{2+}]_i$ and $pH_i$ were measured simultaneously, by co-loading cardiomyocytes with Fura-2 (0.075 $\mu mol\, l^{-1}$, for 10 min) and BCECF (0.8 $\mu mol\, l^{-1}$, for 30 min). ROS were measured separately by loading cardiomyocytes with DHE (5 $\mu mol\, l^{-1}$, for 30 min). Following the loading protocol, cells were resuspended in fresh isolation solution for 15–20 min to allow de-esterification. Cardiomyocytes were then placed in the recording chamber (at room temperature), perfused with normoxic saline and stimulated to contract at a frequency of 0.2 Hz, which is within the normal physiological range of heart-rates in common snapping turtles [21]. After a 10-min stabilization period, the perfusate was switched to an anoxic saline, for 20 min, and cells were subsequently reoxygenated by switching back to the normoxic saline, for a further 10 min (figure 1*a*). $O_2$ was measured continuously throughout the protocol and was undetectable during the anoxic period.

Additionally, we investigated the correlation between external $Ca^{2+}$ concentration and cardiomyocyte length, as a measure of myofilament $Ca^{2+}$-sensitivity, defined here as the relationship between the concentration of free $Ca^{2+}$ ions available to bind to troponin C and the amount of force generated by the cardiomyocyte. Accordingly, myofilament $Ca^{2+}$-sensitivity was determined by perfusing cardiomyocytes with normoxic saline, with increasing levels of extracellular $[Ca^{2+}]$ ($[Ca^{2+}]_e$), from 1 to 5 mM (pH 7.72, at room temperature), while simultaneously measuring relaxed cell length, according to a protocol by Wisløff *et al.* [22].

## (g) Fluorescent imaging

All excitation light waves were filtered with a Nikon T510lpxru dichroic long-pass filter and emitted light waves were collected using HQ535/50 m (Fura-2 and BCECF) and ET585/40 m (DHE) emission filters (Chroma). Fura-2 and BCECF were calibrated using previously published protocols [23–25]. Further details are summarized in the electronic supplementary material.

## (h) Calculations and *statistical analysis*

Statistical analyses and graph plotting were carried out using SIGMAPLOT 13.0 and SPSS 25. Data were tested for equal variances and normality; non-parametric tests were used when data were not normally distributed. Differences in cardiomyocyte volume, width and depth were revealed by Mann–Whitney rank-sum tests, while Student's *t*-tests determined differences in turtle body mass, heart mass, relative heart mass and cardiomyocyte length. Mixed-effects, repeated-measures generalized linear models (GLMs), followed by sequential Sidak *post hoc* tests, were used to determine differences between N21 and H10 cardiomyocyte shortening, $Ca^{2+}$ transients ($\Delta[Ca^{2+}]_i$), normalized $\Delta[Ca^{2+}]_i$, diastolic $[Ca^{2+}]_i$, systolic $[Ca^{2+}]_i$, $pH_i$, ROS production, cardiomyocyte shortening efficiency and times to rise and half-decay. For the GLM analyses, the between-group factor was developmental $O_2$ (N21 versus H10), while time and treatment (normoxia and anoxia) were the within-group factors. A further GLM was used, followed by Sidak *post hoc* tests, to reveal the effect of the developmental $O_2$ and experimental period (normoxia, anoxia or reoxygenation) on cardiomyocyte shortening (the dependent variable), while controlling for $\Delta[Ca^{2+}]_i$ (the covariate). Finally, a mixed-effects, repeated-measures GLM, followed by Sidak *post hoc* tests, was used to reveal the effect of developmental $O_2$ on myofilament $Ca^{2+}$-sensitivity, with relaxed cardiomyocyte length as the dependent variable, while controlling for extracellular $[Ca^{2+}]$ (the covariate). Additionally, regressions were used to fit lines through the data. The $\Delta[Ca^{2+}]_i$ was calculated by subtracting the value of the diastolic

**Table 1.** Morphometric measurements of body mass, heart mass and ventricular cardiomyocyte dimensions from juvenile common snapping turtles that embryonically developed in normoxia (21% $O_2$; N21) or chronic hypoxia (10% $O_2$; H10). Volume, width, depth and length were measured by confocal microscopy analysis. Values are means $\pm$ s.e.m. and statistical significances between the N21 ($n = 15-52$) and H10 ($n = 16$) cohorts are indicated by asterisks when $p \leq 0.05$.

| developmental cohort | body mass (g) | heart mass (mg) | volume ($\mu$l) | width ($\mu$m) | depth ($\mu$m) | length ($\mu$m) |
|---|---|---|---|---|---|---|
| N21 | 285 $\pm$ 20.7 | 747 $\pm$ 56.2 | 7.6 $\pm$ 0.7 | 11 $\pm$ 0.4 | 16.3 $\pm$ 1.1 | 194.5 $\pm$ 5.6 |
| H10 | 292.5 $\pm$ 31.2 | 832.1 $\pm$ 89.4 | 5.6 $\pm$ 0.9* | 13.1 $\pm$ 0.9* | 14.1 $\pm$ 1.6* | 167.5 $\pm$ 10.7* |

$[Ca^{2+}]_i$ from the systolic $[Ca^{2+}]_i$, and ROS values were normalized to the 5-min time-point and described as relative changes in production. All data are presented as means $\pm$ s.e.m. and considered significantly different, when $p \leq 0.05$.

## 3. Results

The mixed-effects, repeated-measures GLMs revealed significant effects of developmental $O_2$ and treatment (anoxia and reoxygenation) on multiple aspects of juvenile turtle cardiomyocyte structure and function. The results are described below, and statistical support for the findings (i.e. test statistics, degrees of freedom and exact $p$-values) are given in electronic supplementary material, tables S2–S7.

### (a) Morphometrics

While the N21 and H10 turtles did not differ in body or heart mass (table 1), there were distinct differences in the size and shape of their ventricular cardiomyocytes. Cardiomyocytes from the H10 versus N21 cohort were significantly smaller, with a reduced length, depth and volume, despite having larger widths (table 1). See electronic supplementary material, table S2, for full details of the statistical analyses.

### (b) Effect of developmental $O_2$ and experimental treatment

The mixed-effects, repeated-measures GLMs revealed a significant effect of developmental $O_2$ on $\Delta[Ca^{2+}]_i$ and cell-shortening efficiency; a significant effect of treatment on cell shortening, $\Delta[Ca^{2+}]_i$, $pH_i$, ROS production, normalized $\Delta[Ca^{2+}]_i$, systolic $[Ca^{2+}]_i$, time to rise and time to half-decay; a significant interaction between normoxia and developmental $O_2$ on the $\Delta[Ca^{2+}]_i$ and cell-shortening efficiency; and a significant interaction between anoxia and developmental $O_2$ on cell shortening, $pH_i$, ROS production and cell-shortening efficiency. See electronic supplementary material, tables S3–S5, for full details of these statistical interactions.

### (c) Normoxic, control conditions

Under control conditions, cell shortening and $pH_i$ of N21 and H10 cardiomyocytes were similar (figure 2a,d), but the average intracellular $Ca^{2+}$ transient ($\Delta[Ca^{2+}]_i$) was more than twice the size in N21 versus H10 cardiomyocytes (figure 2b); this meant the efficiency of shortening (% cell shortening as a function of $\Delta[Ca^{2+}]_i$) was significantly greater in the H10 cohort (figure 2f). The large difference between $\Delta[Ca^{2+}]_i$ values can be explained by lower average N21

versus H10 diastolic $[Ca^{2+}]_i$ levels, but similar average systolic $[Ca^{2+}]_i$; nevertheless, the differences in diastolic $[Ca^{2+}]_i$ did not reach levels of statistical significance (figure 3c,d). Lastly, relative intracellular ROS production did not change during the control period (figure 2e).

### (d) Anoxic perfusion

At the onset of anoxic perfusion, $O_2$ rapidly dropped inside the chamber to undetectable levels (within approx. 54 s), and N21 and H10 contraction declined in association with a reduction in $\Delta[Ca^{2+}]_i$, $pH_i$ and ROS (figure 2a,b,d,e). However, while N21 cell shortening remained depressed throughout the anoxic period, H10 cell shortening recovered to control levels, after 15 min of anoxia (figure 2a), despite a significantly larger acidosis in the H10 cells (figure 2d) and a similar proportional decline in $\Delta[Ca^{2+}]_i$ to the N21 cohort (figure 2c). Therefore, the efficiency of cell shortening during anoxia tended to be higher in the H10 versus N21 cardiomyocytes; this effect was only statistically significant at the 25- and 30-min time-points (figure 2f). In both cell groups, the reduction in the $\Delta[Ca^{2+}]_i$ amplitude (figure 2b) was caused by a progressive increase in diastolic $[Ca^{2+}]_i$ and a decrease in systolic $[Ca^{2+}]_i$, although these values did not reach levels of statistical significance (figure 3c,d). Anoxia also slowed the times to rise and half-decay of both cohorts (table 2). It should be noted that $Ca^{2+}$ transients were not detected during at least one portion of the anoxic challenge, in 9–36% of the 10 N21 and 11 H10 cardiomyocytes tested. ROS production decreased in both N21 and H10 cardiomyocytes, but the proportional decrease was significantly more pronounced in H10 cells (figure 2e).

### (e) Reoxygenation

When $O_2$ was restored (approx. 63 s), N21 and H10 cardiomyocyte shortening recovered to pre-anoxic levels (figure 2a), in association with the restoration of $pH_i$, ROS production (although not to control levels in H10 cells), $[Ca^{2+}]_i$ and the times to rise and half-decay (figure 2b–e and table 2). Overall, most of the examined cardiomyocyte physiological variables returned to control levels, with the exception of diastolic and systolic $[Ca^{2+}]_i$, which remained significantly elevated.

### (f) Cardiomyocyte myofilament $Ca^{2+}$-sensitivity

The observation that H10 cardiomyocyte shortening was either equal to, or larger, than N21 cells, despite smaller $Ca^{2+}$ transients, suggested that the H10 cohort has greater myofilament $Ca^{2+}$-sensitivity. Therefore, we investigated

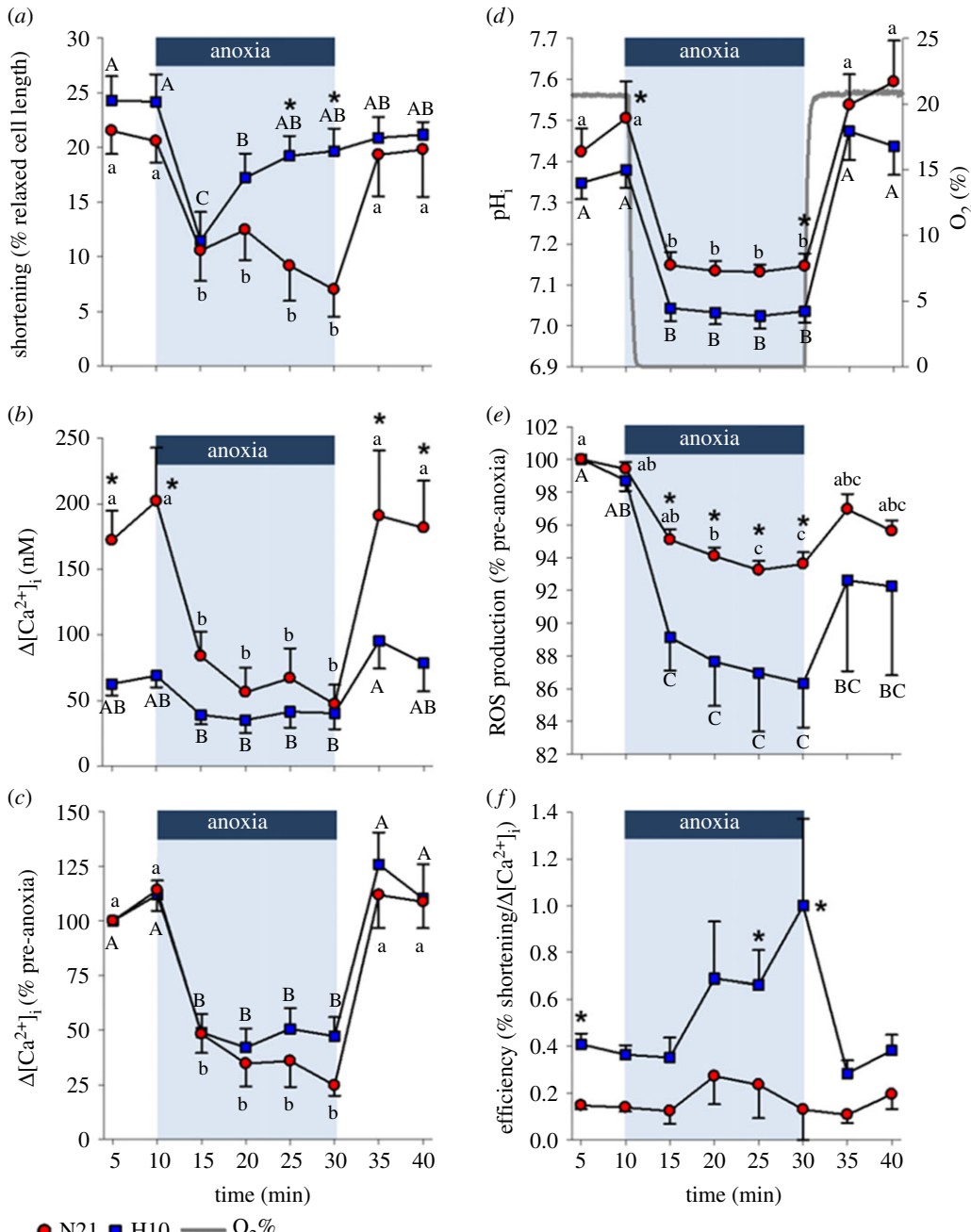

**Figure 2.** Effects of an anoxic challenge on juvenile turtle ventricular cardiomyocyte shortening and ion homeostasis. Cardiomyocytes were isolated from juvenile turtles that embryonically developed in either normoxia (N21) or chronic hypoxia (H10). Cardiomyocytes were exposed to simulated normoxia, anoxia and reoxygenation. (a) Cell shortening was normalized to the relaxed cell length and plotted as a percentage. (b) The absolute and (c) relative intracellular $Ca^{2+}$ transient ($\Delta[Ca^{2+}]_i$). (d) intracellular pH ($pH_i$). (e) Reactive oxygen species (ROS) production; measured with dihydroethidium (DHE), normalized to the 5-min time-point (control) and plotted as a relative change. (f) The efficiency of cell shortening; calculated as the quotient of shortening (% relaxed cell length) divided by $\Delta[Ca^{2+}]_i$ (nM). Statistical significances are indicated by letters, for within-group differences ($^{ab}$N21 and $^{AB}$H10; one-tailed tests used for cell shortening, $pH_i$, and $\Delta[Ca^{2+}]_i$), and asterisks, for between-group differences (*N21 versus H10). Values are means ± s.e.m. and significant when $p \leq 0.05$. Cardiomyocyte shortening, $\Delta[Ca^{2+}]_i$, and $pH_i$ were measured simultaneously ($n = 9-10$ and $10-11$, for N21 and H10, respectively). ROS was measured independently ($n = 8$ and 7, for N21 and H10, respectively). (Online version in colour.)

the correlation between cardiomyocyte contraction and $\Delta[Ca^{2+}]_i$ (figure 4a,b). As expected, regression lines fitted to the data revealed that shortening is positively correlated with $\Delta[Ca^{2+}]_i$, during all three experimental periods, with the exception of N21 cells in anoxia. However, when controlling for $\Delta[Ca^{2+}]_i$, the extent of the shortening was greater in H10 cardiomyocytes, regardless of the experimental period (figure 4a,b). Furthermore, relaxed cardiomyocyte length decreased by a greater proportion in the H10 cohort, at the range of external $Ca^{2+}$ concentrations tested (figure 4c).

Collectively, our data suggest that the myofilaments of H10 cardiomyocytes have a higher sensitivity to $Ca^{2+}$ than N21 cells. See electronic supplementary material, tables S6 and S7, for full details of these statistical interactions.

## 4. Discussion

Numerous studies across different vertebrate taxa show that developmental hypoxia alters cardiac morphology and

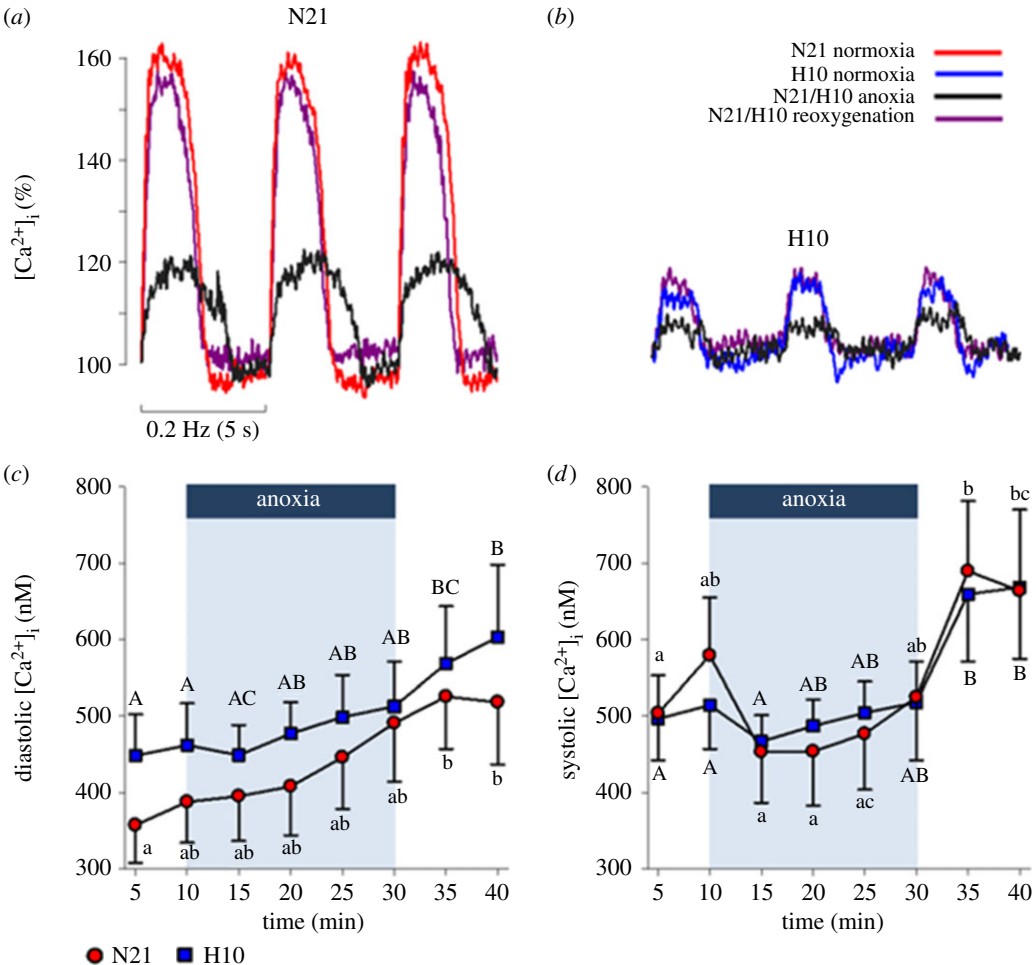

**Figure 3.** Effects of an anoxic challenge on juvenile turtle ventricular cardiomyocyte diastolic and systolic $Ca^{2+}$ levels. Cardiomyocytes were isolated from juvenile turtles that embryonically developed in either normoxia (N21) or chronic hypoxia (H10). Representative traces of normalized $Ca^{2+}$ transients from (*a*) N21 and (*b*) H10 cardiomyocytes demonstrate the effects of normoxia, anoxia and reoxygenation (see legend for colour codes) on transient shape, amplitude and duration. (*c*) Diastolic and (*d*) systolic intracellular $Ca^{2+}$ ($[Ca^{2+}]_i$). Statistical significances are indicated by letters, for within-group differences ([ab]N21 and [AB]H10), and asterisks, for between-group differences (*N21 versus H10). Values are means $\pm$ s.e.m. ($n = 9-10$ and $10-11$, for N21 and H10, respectively) and significant when $p \leq 0.05$. (Online version in colour.)

**Table 2.** The effect of an anoxic challenge on the $Ca^{2+}$-transient kinetics of ventricular cardiomyocytes of juvenile turtles that embryonically developed in either normoxia (21% $O_2$; N21) or chronic hypoxia (10% $O_2$; H10). Values are means $\pm$ s.e.m. and statistical significances are indicated by letters for within-group differences ([ab]N21 and [AB]H10), when $p \leq 0.05$ ($n = 9-10$ and $10-11$, for the N21 and H10 groups, respectively).

| developmental cohort | time to rise (ms) | | | time to half-decay (ms) | | |
|---|---|---|---|---|---|---|
| | normoxia | anoxia | reoxygenation | normoxia | anoxia | reoxygenation |
| N21 | 895 $\pm$ 114.3[a] | 1475 $\pm$ 228.5[b] | 878.6 $\pm$ 102.8[a] | 757.5 $\pm$ 95.1[a] | 1074 $\pm$ 234.4[b] | 839.6 $\pm$ 123.9[ab] |
| H10 | 650.5 $\pm$ 81.5[A] | 1390.3 $\pm$ 209.6[B] | 869.4 $\pm$ 142.7[A] | 567 $\pm$ 69.7[A] | 965.5 $\pm$ 198.5[A] | 764.6 $\pm$ 168[AB] |

function [5,7,13,26–33]. However, very few studies have attempted to characterize these relationships at the cellular level. Here we show, for the first time, that developmental hypoxia alters juvenile turtle cardiomyocyte morphology, ion homoeostasis and the cellular responses to anoxia and reoxygenation. Importantly, our results suggest that exposure to hypoxia during development improves cardiomyocyte anoxia-tolerance of juvenile snapping turtles, which appears to be supported by enhanced cell-shortening efficiency (figure 2*f*), higher myofilament $Ca^{2+}$-sensitivity (figure 4) and a superior ability to suppress ROS production (figure 2*e*).

## (a) Developmental hypoxia alters cardiomyocyte structure and function

Previous work has shown developmental hypoxia is associated with cardiac enlargement in embryonic snapping turtles [11], but this phenotype does not persist into juvenile life [14]. In this study, we found that absolute body mass, absolute heart mass and relative heart mass were similar between the developmental cohorts, but that cardiomyocyte length, depth and volume were smaller in juvenile turtles previously exposed to developmental hypoxia (table 1).

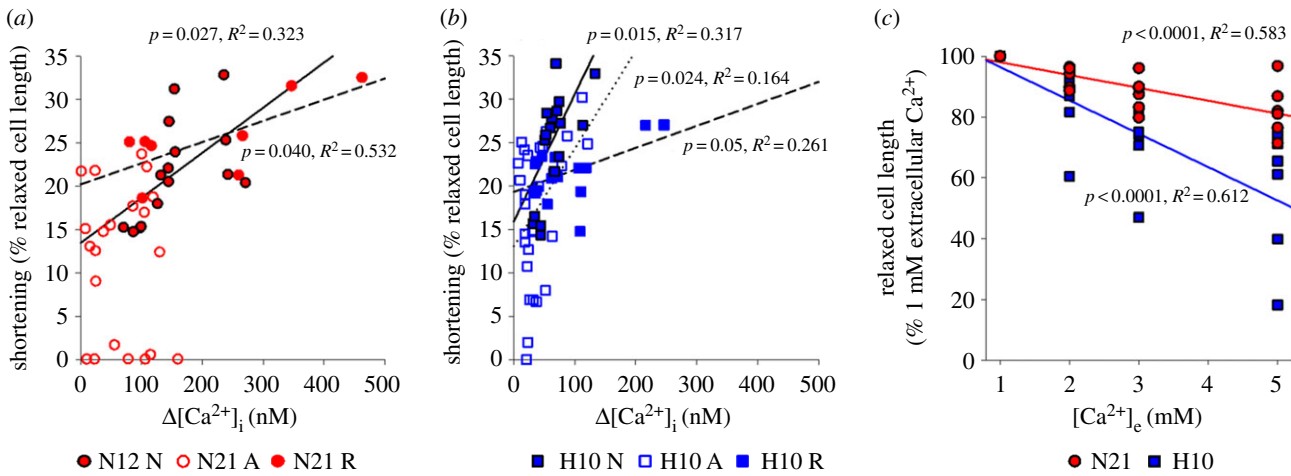

**Figure 4.** The effect of developmental hypoxia and experimental treatment on myofilament $Ca^{2+}$-sensitivity. (a,b) Correlations between the calcium transient ($\Delta[Ca^{2+}]_i$) and cell shortening, in cardiomyocytes from juvenile turtles that embryonically developed in either normoxia (N21) or chronic hypoxia (H10). Plotted data represent normoxia (N), anoxia (A) and reoxygenation (R); N21 n-values = 26, 52 and 26 and H10 n-values = 22, 44 and 22, for normoxia, anoxia and reoxygenation, respectively. (c) Correlation between extracellular $Ca^{2+}$ ($[Ca^{2+}]_e$) and relaxed cell length ($n = 32$ and 24, for N21 and H10, respectively). (Online version in colour.)

Therefore, cardiac hyperplasia, possibly combined with an increase in the mass of structural components of the heart (e.g. fibroblasts or collagen), is likely to accompany the reduced cardiomyocyte size in the H10 cohort.

Alongside differences in cell morphology, developmental hypoxia also affected the functional properties of turtle cardiomyocytes. The most striking difference was the remarkably smaller H10 calcium transient ($\Delta[Ca^{2+}]_i$), compared with N21 cardiomyocytes (figure 2b). Differences in cardiomyocyte $\Delta[Ca^{2+}]_i$ can be explained by numerous factors, including $Ca^{2+}$-influx pathways (e.g. L-type $Ca^{2+}$ channel or the sarcoplasmic reticulum), efflux pathways (e.g. the $Na^+/Ca^{2+}$-exchanger) or buffering capacity (e.g. calmodulin). While we did not investigate these $Ca^{2+}$-cycling mechanisms here, we found that H10 cardiomyocytes could still develop the same level of shortening as the N21 cells, despite the smaller $\Delta[Ca^{2+}]_i$. Our findings suggest that these differences can partly be explained by improved myofilament $Ca^{2+}$-sensitivity in the H10 cohort (figure 4). High $Ca^{2+}$-sensitivity of ventricular myofibrils is a general characteristic of ectothermic vertebrates [34,35] and is suggested to aid in the maintenance of contractile activity, during an environmental stress, such as low temperatures or hypoxia. In line with our results, developmental hypoxia at high altitude also improves $Ca^{2+}$-sensitivity in the ventricular muscle of fetal sheep [36]. Interestingly, a recent study demonstrated that the gene expression of cardiac troponin is epigenetically modulated by DNA methylation during development [37]. Therefore, developmental hypoxia might enhance myofilament $Ca^{2+}$-sensitivity by modulating the methylation patterns of genes that code for troponin isoforms; further work is necessary to test this intriguing hypothesis.

In addition to contributing towards the maintenance of contractility, a greater myofilament $Ca^{2+}$-sensitivity might also have implications for energy expenditure, when considering the higher shortening efficiency of H10 cardiomyocytes (figure 2f). For example, $Ca^{2+}$-cycling is a major consumer of ATP, on account of the high energetic demand of $Ca^{2+}$-removal pathways, such as the sarcoplasmic reticulum $Ca^{2+}$-ATPase and the $Na^+/Ca^{2+}$ exchanger, which is indirectly

dependent on the $Na^+/K^+$-ATPase [38]. Thus, it is possible that increasing myofilament $Ca^{2+}$-sensitivity will decrease ATP demand, because the cell would require less $Ca^{2+}$ to produce the same level of contractile force. Indeed, there is evidence that increasing myofilament $Ca^{2+}$-sensitivity improves the energy economy of a cell, which is characterized by stronger twitch tension (that does not affect times to peak or relaxation), higher turnover of crossbridge formation and faster $Ca^{2+}$-cycling [39]. With respect to our findings, it would, therefore, be interesting to determine the metabolic cost of contractility in the N21 and H10 cohorts.

## (b) Effects of simulated anoxia on cardiomyocyte anoxia-tolerance

N21 and H10 cardiomyocyte shortening was dramatically impaired within the first 5 min of the anoxic challenge. Consistent with this finding, in vivo and in vitro turtle studies show that anoxia reduces ventricular pressure and cardiac output [40,41]. Our results suggest that depressed cell shortening is partly due to two factors: a diminished $\Delta[Ca^{2+}]_i$, which reduces myofilament crossbridge formation, and a profound intracellular acidosis, brought on by a respiratory and a metabolic acidosis, which reduces myofilament $Ca^{2+}$-sensitivity [42]. However, while N21 cardiomyocyte shortening remained depressed, H10 cell shortening rebounded to pre-anoxic levels and cell-shortening efficiency remained unchanged, despite a persistent reduction in $\Delta[Ca^{2+}]_i$ and a lower $pH_i$ than N21 cardiomyocytes. Evidently, prior exposure to developmental hypoxia improved cardiomyocyte anoxia-tolerance in juvenile turtles.

Developmental programming of stress-tolerant phenotypes is well documented in teleost fish. Adult zebrafish (Danio rerio) exposed to hypoxia produce offspring that are more hypoxia-tolerant [43], and zebrafish larvae from embryos that developed in hypoxia have a lower critical $O_2$ tension, compared with their normoxic counterparts [44]. Similarly, adult killifish (Fundulus heteroclitus) and white suckerfish (Catostomus commersonii) subjected to polluted environments produce offspring that are more resistant to

the specific pollutant [45,46]. Granted, there are numerous examples of pathological programming in both ectothermic and endothermic vertebrates subjected to hypoxia (for reviews, see [5,6,47]). Nevertheless, our results support the concept that early exposure to hypoxia might prime some ectothermic vertebrates to better cope with hypoxic environments in adulthood, particularly those that frequently encounter hypoxia throughout their lives.

Given that $\Delta[Ca^{2+}]_i$ remained relatively stable during anoxia, the mechanism underlying the rebound in cell shortening of H10 cardiomyocytes must be $Ca^{2+}$-independent (figure 2a,b). Consistent with the hypothesis that developmental hypoxia enhances myofilament $Ca^{2+}$-sensitivity, cell shortening in N21 and H10 cardiomyocytes was similar within the first 5 min of anoxia, despite the lower intracellular pH of the H10 cohort. However, the time course of the rebound in H10 cell shortening suggests separate mechanisms were activated to recover contractile function. One possibility is the suppression of ROS production. Low $O_2$ tensions are associated with excessive ROS production, which is known to reduce cardiac force production, by directly modifying sarcomere proteins and reducing myofilament $Ca^{2+}$-sensitivity [48–50]. Our results are in line with several studies that have shown turtles suppress ROS production in the brain and heart during anoxic exposure [49,51]. However, the magnitude of the relative suppression of ROS production was significantly greater in the H10 cohort, which might contribute to the recovery of cell shortening. Collectively, these results suggest developmental hypoxia alters pathways involved in ROS management that might protect the heart against oxidative stress. Future research should be devoted to identifying the mechanisms underlying the programming of oxidative-stress pathways in turtles.

## (c) Effects of simulated reoxygenation on cardiomyocyte function

Although anoxia can produce extensive tissue damage in mammals, the greatest risk to cell survival occurs after reoxygenation, when the mitochondria produce a surge of ROS that can trigger apoptotic and necrotic cell death [16,52]. By contrast, our study, and several others [51,53,54], have shown that reoxygenation causes no damage to the turtle heart. Here we demonstrate that N21 and H10 cell

shortening, $pH_i$, $\Delta[Ca^{2+}]_i$ and ROS production all recovered to intermediate or pre-anoxic levels within the first 5 min of reoxygenation, despite the persistent increases in diastolic and systolic $[Ca^{2+}]_i$. These results suggest the turtle heart can withstand reoxygenation without apparent cellular injury, irrespective of developmental $O_2$ tension.

## 5. Perspectives

North American freshwater turtles experience prolonged periods of $O_2$ deprivation throughout their lifetime [3,9]. We have shown evidence that cardiac anoxia-tolerance of turtles can be improved by early exposure to hypoxia during development. Mechanistically, our data suggest the improvement in cardiac anoxia-tolerance is supported by greater myofilament $Ca^{2+}$-sensitivity and a superior ability to suppress ROS production. From an ecological perspective, these modifications might be beneficial to turtles and possibly other ectotherms, when they exploit hypoxic environments.

Ethics. Permission to collect snapping turtle eggs granted by State of Minnesota Department of Natural Resources, Division of Fish and Wildlife, Section of Fisheries permit no. 21232.

Turtle husbandry and experimental procedures were carried out in accordance with University of Manchester handling protocols, which adhere to the United Kingdom Home Office legislation. Project licence to perform the experiments was granted to G.L.J.G.

Data accessibility. All data collected, as part of this work, are directly presented in the text of the manuscript and can be found in the electronic supplementary material.

Authors' contributions. Conceptualization: I.M.R., G.L.J.G.; Methodology: I.M.R., G.L.J.G., H.A.S., D.A.C.; Formal analysis: I.M.R., G.L.J.G.; Investigation: I.M.R., G.L.J.G., H.M., A.B.; Resources: G.L.J.G.; Writing—original draft: I.M.R.; writing—review and editing: I.M.R., G.L.J.G., H.A.S., D.A.C.; Visualization: I.M.R.; Supervision: G.L.J.G.; Project administration: G.L.J.G.; Funding acquisition: G.L.J.G.

Competing interests. We declare we have no competing interests.

Funding. This study was funded by a New Investigator Grant awarded to G.L.J.G. by the Biotechnology and Biological Sciences Research Council (BBSRC grant no. BB/N005740/1).

Acknowledgements. We thank the staff of the Biological Services Facility, at the University of Manchester (UoM), for their help with turtle husbandry. We thank Prof Andrew Trafford (UoM) for experimental advice and the use of his equipment and assistance from Jessica Caldwell (UoM) with confocal microscopy and Peter Ceuppens (UoM) with our statistical analyses.

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
