## [Reviewer comments · Proceedings of the Royal Society B: Biological Sciences]

Review History

RSPB-2019-0259.R0 (Original submission)

Review form: Reviewer 1

Recommendation

Major revision is needed (please make suggestions in comments)

Scientific importance: Is the manuscript an original and important contribution to its field?

Excellent

General interest: Is the paper of sufficient general interest?

Good

Quality of the paper: Is the overall quality of the paper suitable?

Good

Is the length of the paper justified?

Yes

Should the paper be seen by a specialist statistical reviewer?

Yes

Do you have any concerns about statistical analyses in this paper? If so, please specify them explicitly in your report.

Yes

It is a condition of publication that authors make their supporting data, code and materials available - either as supplementary material or hosted in an external repository. Please rate, if applicable, the supporting data on the following criteria.

Is it accessible?

Yes

Is it clear?

Yes

Is it adequate?

Yes

Do you have any ethical concerns with this paper?

No

Comments to the Author

In this study, Ruhr and colleagues investigated the effect of hypoxic developmental programming (10 vs 21% O₂ exposure in ovo) on anoxia tolerance in cardiomyocytes of snapping turtles. The manuscript is eminently readable (the introduction is particularly beautifully written) and the authors must also be commended on figures that are pleasing to the eye. The methods are well described and I trust that the experiments were expertly performed.

The major conclusion (i.e. summed in the final paragraph, lines 276-278) is that the hypoxia programmed turtles have more tolerant hearts because they had greater myofilament calcium sensitivity and a better ability to suppress ROS. Unfortunately, I have both methodological and statistical concerns for the validity of these conclusions that I believe should be addressed.

Major comments:

The authors claim to provide 'mechanistic insight' (line 68), but I believe the study is actually rather descriptive; although they are thorough in their descriptions, which include a variety of measurements, this doesn't equate to 'mechanistic'. This is even acknowledged by the authors on line 215-216 ('we did not investigate these mechanisms here'). I think this is a problem because some of the results, such as the huge (threefold) difference in calcium transient, are so staggering. Such findings would be more credible if the underlying mechanism for the differences could be shown (e.g. using a pharmacological approach to dissect calcium handling in the two groups and show how they are different).

Initially, I had trouble finding the data on myofilament Ca²⁺ sensitivity because it has been relegated to the 'Supplementary Results'. Given that this is presented as an important finding, I am unsure why it is hidden like this if the authors have full confidence in it.

At first sight, it appears reasonable that the cardiomyocytes of hypoxia programmed animals could have greater myofilament calcium sensitivity because they exhibited similar, or even greater, contractility (cell shortening) despite having a Ca²⁺ transient that was threefold smaller than in the cells from the normoxic turtles.

Unfortunately, upon examining Figure 4, this quickly comes into doubt- the hypoxia tolerant animals had similar systolic Ca²⁺ but much higher diastolic Ca²⁺- if their myofilaments were more sensitive to Ca²⁺ it could, in my mind, spell serious problems for cardiac relaxation (diastolic dysfunction), which is inconsistent with their apparently normal contractile phenotype. I therefore highly doubt this explanation, based on the data that is presented.

If the authors remain compelled that this supports their 'mechanism', there are a number of established methods for investigating myofilament Ca²⁺ sensitivity directly (skinned preparations, steady-state contractures). These rely on steady-state measurements of force when the myofilaments are exposed to varying Ca²⁺. But these techniques were not employed in the present study. Instead, they resort to very tenuous reasoning in the Supplementary Results that quickly becomes circular (basically, as far as I can interpret it, the myofilaments must have been more sensitive because they have smaller calcium transients but similar force. This is not the equivalent to demonstrating increased calcium sensitivity).

The statistics are not thoroughly described and there are a few points where I would like to better understand how they have chosen their tests.

Figure 3: There are two-factors: turtle programming (H or N) and acute anoxia exposure of the cells (i.e. time). The authors describe performing, separately, as far as I can see: "one-way, repeated-measures ANOVAs, followed by Holm-Sidak post-hoc tests, for within-group comparisons and individual Student's t-tests or Mann-Whitney rank-sum tests for between-group comparisons."

Why did they choose to do this as opposed to analyzing all of the data together in a two-way ANOVA, or, to avoid the issue of incomplete data (some points are missing when the cells died), a mixed effects linear model? This would provide a much better insight into the data, complete with interactions.

The biggest issue, I believe, with the present analysis is the use of separate t-tests for the between group comparisons- this seems inappropriate to me without a correction for multiple comparisons (e.g. Bonferroni correction).

Out of curiosity, I quickly analysed the raw data for ROS production with a linear mixed effects analysis followed by Bonferroni multiple comparisons (using GraphPad Prism) and believe there may be no significantly differences between the programming groups. Can the authors justify the individual t-tests?

As a lesser issue, when were the Student's t-tests or Mann-Whitney rank-sum tests used? Were the data tested for normality (this isn't stated)? If so, I think it is important to state which tests were used for which data.

Line 167-168: the authors say that ROS production was similar under (starting) control conditions, but all of the cells were normalised to 100% at the start of the trial, so of course they appeared similar during this initial period of normoxia?

This naivety potentially highlights a more substantial concern that baseline ROS production could actually have been different at the start and we have no way of knowing, is there any way of truly indicating that the ROS production was similar at the start?

To me it is unclear what condition the protocol, particularly with these saline solutions, is trying to simulate and how it is relevant for this species. The cardiomyocytes were exposed to a rapid and extreme decrease in HCO₃ from 35 to 15 mM which, combined with the increase in CO₂, causes a huge fall in pH (7.7 to 6.8). The authors state that the salines were designed based on in vivo studies from turtles exposed to normoxia and anoxia.

The first, small, issue is that they only cite work performed on *Chrysemys*, which is quite distantly related from *Chelydra*. A relevant paper to consult would be

- Reese, S.A., Jackson, D.C., Ultsch, G.R., 2015. The Physiology of Overwintering in a Turtle That Occupies Multiple Habitats, the Common Snapping Turtle (*Chelydra serpentina*). *Physiological and Biochemical Zoology* 75, 432–438. doi:10.1086/342802

Which as far as I can see has not been cited. This, nonetheless, shows relatively similar changes to *Chrysemys*.

However, more pressingly, most of these *in vivo* studies (included those already provided by the authors) are on the effects of chronic (typically days) of anoxia, how is this confluent with their acute 30 min exposure of the cardiomyocytes?

Another paper that I believe has been overlooked by the authors, that is specific to their study species, is:

- Frische, S., Fago, A., Altimiras, J., 2000. Respiratory responses to short term hypoxia in the snapping turtle, *Chelydra serpentina*. *Comp. Biochem. Physiol., Part A Mol. Integr. Physiol.* 126, 223–231. doi:10.1016/S1095-6433(00)00201-4

Here it is shown that when this species is exposed to short term (aerial) hypoxia, ventilation increases meaning that plasma pH increases (opposite to the huge decrease in the protocol of the present paper). I think they need to better explain what sort of hypoxia they are interested in.

As a distinct but related point, for the purposes of clearly understanding the response of the heart and providing comparisons to previous studies (some of which have used anoxia but no acidosis), I think it is important to emphasise in the main text that the present study explored combined anoxia and hypercapnic acidosis- the latter is hardly acknowledged. For example, on line 232 the authors report a ‘profound intracellular acidosis’, but this is not surprising in comparison to the even more profound extracellular acidosis.

Minor comments:

For me, the terms N21 and H10 are something of a pleonasm. I don’t think it is necessary to consistently reinforce that normoxia is 21% oxygen once it has been defined as such. Also, if, as described in the second paragraph of the introduction, eggs in the wild are often exposed to hypoxia, is ‘normoxic’ really ‘normal’ for these animals?

Line 133- It isn’t patently clear what temperature the cells were exposed to in the closed cell bath (recording chamber)? I can see that they were loaded at room temperature, is the assumption then that they were also studied at that temperature?

Line 134- how was this stimulation frequency chosen? It (i.e. 12 beats per minute) is approximately half the heart rate value reported in the two Wearing et al. (2016 and 2017) papers cited by the authors, although I would again encourage the authors to refer to Frische et al. (2000, CBP), where lower heart rates were observed.

Line 191- was systolic Ca²⁺ not also still elevated?

Line 269- a similar issue to above, you say intracellular Ca²⁺ returned to intermediate on pre-anoxic levels, but whilst this was true for the transient, it cannot be said for the systolic or

diastolic concentrations per se.

Review form: Reviewer 2

Recommendation

Accept with minor revision (please list in comments)

Scientific importance: Is the manuscript an original and important contribution to its field?

Excellent

General interest: Is the paper of sufficient general interest?

Excellent

Quality of the paper: Is the overall quality of the paper suitable?

Excellent

Is the length of the paper justified?

Yes

Should the paper be seen by a specialist statistical reviewer?

No

Do you have any concerns about statistical analyses in this paper? If so, please specify them explicitly in your report.

No

It is a condition of publication that authors make their supporting data, code and materials available - either as supplementary material or hosted in an external repository. Please rate, if applicable, the supporting data on the following criteria.

Is it accessible?

Yes

Is it clear?

Yes

Is it adequate?

Yes

Do you have any ethical concerns with this paper?

No

Comments to the Author

This manuscript by Ruhr et al reports an interesting study on how exposure to hypoxia during development protect cardiomyocytes from hypoxic stress later in life. The authors chose a well-suited species, the snapping turtle, a species highly tolerant of severe hypoxia, and appropriate combination of cutting edge techniques for monitoring cell morphology, shortening, intracellular Ca²⁺ and pH as well as ROS production of intact, isolated cardiomyocytes in a normoxia-anoxia-recovery protocol. The manuscript is very well written and the experiments have been carefully conducted. The conclusions that cardiomyocytes from hypoxic developing turtles contract more

efficiently and produce less ROS are highly valuable and improve our understanding on the mechanisms controlling hypoxia tolerance of the vertebrate heart. I have only a few suggestions for the authors to consider, to improve this manuscript further.

Line 165: calcium transient kinetics. Make clear if these refer to traces A and B in Figure 4. In the same Figure 4, panels E and F, please eliminate the line connecting the data points since there are no intermediate conditions in between treatments.

Line 213 and 249: please add the reference to the figure in question, to help the reader.

Line 218: please explain how is Ca²⁺ sensitivity defined. Is it the Ca²⁺ concentration necessary to achieve a given contraction?

In Figure 2, please add a color-code legend explaining that red bars refer to N21 and blue bars to H10

In Figure 3, the much higher efficiency of contraction per unit of Ca²⁺ gradient of N10 turtles is a remarkable result in the context of energy-saving that however is not much discussed as it deserves. Also, given the role of Ca²⁺ as trigger of hypoxic cell death it would be interesting to know whether data of Figure 4 C or D could indicate (or perhaps not?) channel arrest, as defined by Hochachka? It seems that Ca²⁺ is leaking into cells over time (Fig. 4C)? I wonder whether the authors could expand more on these intriguing issues.

Decision letter (RSPB-2019-0259.R0)

07-Mar-2019

Dear Dr Ruhr:

I am writing to inform you that your manuscript RSPB-2019-0259 entitled "Developmental plasticity of cardiac anoxia-tolerance in juvenile common snapping turtles (*Chelydra serpentina*)" has, in its current form, been rejected for publication in Proceedings B.

This action has been taken on the advice of referees, who have recommended that new analyses and possible further experimental workday be necessary. With this in mind we would be willing to consider a resubmission, provided the comments of the referees are fully addressed. However please note that this is not a provisional acceptance.

To upload a resubmitted manuscript, log into <http://mc.manuscriptcentral.com/prsb> and enter

your Author Centre, where you will find your manuscript title listed under "Manuscripts with Decisions." Under "Actions," click on "Create a Resubmission." Please be sure to indicate in your cover letter that it is a resubmission, and supply the previous reference number.

Sincerely,
Victoria Braithwaite

=====
Professor V A Braithwaite
Proceedings B
mailto: proceedingsb@royalsociety.org

====
Associate Editor

Comments to Author:

Both reviewers agree, as do I, that the MS is interesting, well written and provides valuable new information. They do however provide some suggestions to improve the MS, which should be considered in a revised version. In particular, I agree with Reviewer One that the statistical analyses need to be much more clearly described, and statistical support for the findings needs to be presented in the text. Currently a brief description of the statistical methodology and results are presented in the figure legends, rather than the methods and results. I think a one-way RMANOVA is appropriate for the data presented in Figs 3 & 4, but a more detailed description in the methods (i.e. indicating time is the repeat and hypoxia is the factor) would help. However, I don't understand the use of post hoc tests (my understanding is post hoc tests are not available for RMANOVA) and the between-group comparisons (this comparison can be provided by the factor, and as pair-wise comparisons needs a multiple comparison correction). Please provide statistical support for your findings in the results i.e. test statistics, model and error degrees of freedom, exact p value).

==
Reviewers' Comments to Author:

Referee: 1

In this study, Ruhr and colleagues investigated the effect of hypoxic developmental programming (10 vs 21% O₂ exposure in ovo) on anoxia tolerance in cardiomyocytes of snapping turtles. The manuscript is eminently readable (the introduction is particularly beautifully written) and the authors must also be commended on figures that are pleasing to the eye. The methods are well described and I trust that the experiments were expertly performed.

The major conclusion (i.e. summed in the final paragraph, lines 276-278) is that the hypoxia programmed turtles have more tolerant hearts because they had greater myofilament calcium sensitivity and a better ability to suppress ROS. Unfortunately, I have both methodological and statistical concerns for the validity of these conclusions that I believe should be addressed.

Major comments:

The authors claim to provide 'mechanistic insight' (line 68), but I believe the study is actually rather descriptive; although they are thorough in their descriptions, which include a variety of measurements, this doesn't equate to 'mechanistic'. This is even acknowledged by the authors on line 215-216 ('we did not investigate these mechanisms here'). I think this is a problem because

some of the results, such as the huge (threefold) difference in calcium transient, are so staggering. Such findings would be more credible if the underlying mechanism for the differences could be shown (e.g. using a pharmacological approach to dissect calcium handling in the two groups and show how they are different).

Initially, I had trouble finding the data on myofilament Ca^{2+} sensitivity because it has been relegated to the 'Supplementary Results'. Given that this is presented as an important finding, I am unsure why it is hidden like this if the authors have full confidence in it.

At first sight, it appears reasonable that the cardiomyocytes of hypoxia programmed animals could have greater myofilament calcium sensitivity because they exhibited similar, or even greater, contractility (cell shortening) despite having a Ca^{2+} transient that was threefold smaller than in the cells from the normoxic turtles.

Unfortunately, upon examining Figure 4, this quickly comes into doubt- the hypoxia tolerant animals had similar systolic Ca^{2+} but much higher diastolic Ca^{2+} - if their myofilaments were more sensitive to Ca^{2+} it could, in my mind, spell serious problems for cardiac relaxation (diastolic dysfunction), which is inconsistent with their apparently normal contractile phenotype. I therefore highly doubt this explanation, based on the data that is presented.

If the authors remain compelled that this supports their 'mechanism', there are a number of established methods for investigating myofilament Ca^{2+} sensitivity directly (skinned preparations, steady-state contractures). These rely on steady-state measurements of force when the myofilaments are exposed to varying Ca^{2+} . But these techniques were not employed in the present study. Instead, they resort to very tenuous reasoning in the Supplementary Results that quickly becomes circular (basically, as far as I can interpret it, the myofilaments must have been more sensitive because they have smaller calcium transients but similar force. This is not the equivalent to demonstrating increased calcium sensitivity).

The statistics are not thoroughly described and there are a few points where I would like to better understand how they have chosen their tests.

Figure 3: There are two-factors: turtle programming (H or N) and acute anoxia exposure of the cells (i.e. time). The authors describe performing, separately, as far as I can see: "one-way, repeated-measures ANOVAs, followed by Holm-Sidak post-hoc tests, for within-group comparisons and individual Student's t-tests or Mann-Whitney rank-sum tests for between-group comparisons."

Why did they choose to do this as opposed to analyzing all of the data together in a two-way ANOVA, or, to avoid the issue of incomplete data (some points are missing when the cells died), a mixed effects linear model? This would provide a much better insight into the data, complete with interactions.

The biggest issue, I believe, with the present analysis is the use of separate t-tests for the between group comparisons- this seems inappropriate to me without a correction for multiple comparisons (e.g. Bonferroni correction).

Out of curiosity, I quickly analysed the raw data for ROS production with a linear mixed effects analysis followed by Bonferroni multiple comparisons (using GraphPad Prism) and believe there may be no significant differences between the programming groups. Can the authors justify the individual t-tests?

As a lesser issue, when were the Student's t-tests or Mann-Whitney rank-sum tests used? Were the data tested for normality (this isn't stated)? If so, I think it is important to state which tests were used for which data.

Line 167-168: the authors say that ROS production was similar under (starting) control conditions, but all of the cells were normalised to 100% at the start of the trial, so of course they appeared similar during this initial period of normoxia?

This naivety potentially highlights a more substantial concern that baseline ROS production could actually have been different at the start and we have no way of knowing, is there any way of truly indicating that the ROS production was similar at the start?

To me it is unclear what condition the protocol, particularly with these saline solutions, is trying to simulate and how it is relevant for this species. The cardiomyocytes were exposed to a rapid and extreme decrease in HCO₃ from 35 to 15 mM which, combined with the increase in CO₂, causes a huge fall in pH (7.7 to 6.8). The authors state that the salines were designed based on in vivo studies from turtles exposed to normoxia and anoxia.

The first, small, issue is that they only cite work performed on *Chrysemys*, which is quite distantly related from *Chelydra*. A relevant paper to consult would be

- Reese, S.A., Jackson, D.C., Ultsch, G.R., 2015. The Physiology of Overwintering in a Turtle That Occupies Multiple Habitats, the Common Snapping Turtle (*Chelydra serpentina*). *Physiological and Biochemical Zoology* 75, 432–438. doi:10.1086/342802

Which as far as I can see has not been cited. This, nonetheless, shows relatively similar changes to *Chrysemys*.

However, more pressingly, most of these in vivo studies (included those already provided by the authors) are on the effects of chronic (typically days) of anoxia, how is this confluent with their acute 30 min exposure of the cardiomyocytes?

Another paper that I believe has been overlooked by the authors, that is specific to their study species, is:

- Frische, S., Fago, A., Altimiras, J., 2000. Respiratory responses to short term hypoxia in the snapping turtle, *Chelydra serpentina*. *Comp. Biochem. Physiol., Part A Mol. Integr. Physiol.* 126, 223–231. doi:10.1016/S1095-6433(00)00201-4

Here it is shown that when this species is exposed to short term (aerial) hypoxia, ventilation increases meaning that plasma pH increases (opposite to the huge decrease in the protocol of the present paper). I think they need to better explain what sort of hypoxia they are interested in.

As a distinct but related point, for the purposes of clearly understanding the response of the heart and providing comparisons to previous studies (some of which have used anoxia but no acidosis), I think it is important to emphasise in the main text that the present study explored combined anoxia and hypercapnic acidosis- the latter is hardly acknowledged. For example, on line 232 the authors report a 'profound intracellular acidosis', but this is not surprising in comparison to the even more profound extracellular acidosis.

Minor comments:

For me, the terms N21 and H10 are something of a pleonasm. I don't think it is necessary to consistently reinforce that normoxia is 21% oxygen once it has been defined as such. Also, if, as described in the second paragraph of the introduction, eggs in the wild are often exposed to hypoxia, is 'normoxic' really 'normal' for these animals?

Line 133- It isn't patently clear what temperature the cells were exposed to in the closed cell bath (recording chamber)? I can see that they were loaded at room temperature, is the assumption then that they were also studied at that temperature?

Line 134- how was this stimulation frequency chosen? It (i.e. 12 beats per minute) is approximately half the heart rate value reported in the two Wearing et al. (2016 and 2017) papers cited by the authors, although I would again encourage the authors to refer to Frische et al. (2000, CBP), where lower heart rates were observed.

Line 191- was systolic Ca²⁺ not also still elevated?

Line 269- a similar issue to above, you say intracellular Ca²⁺ returned to intermediate on pre-anoxic levels, but whilst this was true for the transient, it cannot be said for the systolic or diastolic concentrations per se.

==

Referee: 2

This manuscript by Ruhr et al reports an interesting study on how exposure to hypoxia during development protect cardiomyocytes from hypoxic stress later in life. The authors chose a well-suited species, the snapping turtle, a species highly tolerant of severe hypoxia, and appropriate combination of cutting edge techniques for monitoring cell morphology, shortening, intracellular Ca²⁺ and pH as well as ROS production of intact, isolated cardiomyocytes in a normoxia-anoxia-recovery protocol. The manuscript is very well written and the experiments have been carefully conducted. The conclusions that cardiomyocytes from hypoxic developing turtles contract more efficiently and produce less ROS are highly valuable and improve our understanding on the mechanisms controlling hypoxia tolerance of the vertebrate heart. I have only a few suggestions for the authors to consider, to improve this manuscript further.

Line 165: calcium transient kinetics. Make clear if these refer to traces A and B in Figure 4. In the same Figure 4, panels E and F, please eliminate the line connecting the data points since there are no intermediate conditions in between treatments.

Line 213 and 249: please add the reference to the figure in question, to help the reader.

Line 218: please explain how is Ca²⁺ sensitivity defined. Is it the Ca²⁺ concentration necessary to achieve a given contraction?

In Figure 2, please add a color-code legend explaining that red bars refer to N21 and blue bars to H10

In Figure 3, the much higher efficiency of contraction per unit of Ca²⁺ gradient of N10 turtles is a remarkable result in the context of energy-saving that however is not much discussed as it deserves. Also, given the role of Ca²⁺ as trigger of hypoxic cell death it would be interesting to know whether data of Figure 4 C or D could indicate (or perhaps not?) channel arrest, as defined by Hochachka? It seems that Ca²⁺ is leaking into cells over time (Fig. 4C)? I wonder whether the authors could expand more on these intriguing issues.

Author's Response to Decision Letter for (RSPB-2019-0259.R0)

See Appendix A.

RSPB-2019-1072.R0

Review form: Reviewer 1

Recommendation

Accept as is

Scientific importance: Is the manuscript an original and important contribution to its field?

Good

General interest: Is the paper of sufficient general interest?

Good

Quality of the paper: Is the overall quality of the paper suitable?

Excellent

Is the length of the paper justified?

Yes

Should the paper be seen by a specialist statistical reviewer?

No

Do you have any concerns about statistical analyses in this paper? If so, please specify them explicitly in your report.

No

It is a condition of publication that authors make their supporting data, code and materials available - either as supplementary material or hosted in an external repository. Please rate, if applicable, the supporting data on the following criteria.

Is it accessible?

Yes

Is it clear?

Yes

Is it adequate?

Yes

Do you have any ethical concerns with this paper?

No

Comments to the Author

Ruhr et al. provide a comprehensive insight into cardiomyocyte physiology of snapping turtles following normoxic or hypoxic embryonic incubation. I enjoyed reviewing this manuscript a second time and am pleased to see that my original concerns have been comprehensively addressed. The new statistical approach is much more appropriate and the new data on myofilament calcium sensitivity is more convincing. I wish to add no further comments.

Decision letter (RSPB-2019-1072.R0)

05-Jun-2019

Dear Dr Ruhr

We are pleased to inform you that your Review manuscript RSPB-2019-1072 entitled "Developmental plasticity of cardiac anoxia-tolerance in juvenile common snapping turtles (*Chelydra serpentina*)" has been accepted for publication in Proceedings B.

The referee does not recommend any further changes. Therefore, please proof-read your manuscript carefully and upload your final files for publication. Because the schedule for publication is very tight, it is a condition of publication that you submit the revised version of your manuscript within 7 days. If you do not think you will be able to meet this date please let me know immediately.

To upload your manuscript, log into <http://mc.manuscriptcentral.com/prsb> and enter your Author Centre, where you will find your manuscript title listed under "Manuscripts with Decisions." Under "Actions," click on "Create a Revision." Your manuscript number has been appended to denote a revision.

You will be unable to make your revisions on the originally submitted version of the manuscript. Instead, upload a new version through your Author Centre.

- 1) A text file of the manuscript (doc, txt, rtf or tex), including the references, tables (including captions) and figure captions. Please remove any tracked changes from the text before submission. PDF files are not an accepted format for the "Main Document".
- 2) A separate electronic file of each figure (tiff, EPS or print-quality PDF preferred). The format should be produced directly from original creation package, or original software format. Please note that PowerPoint files are not accepted.

- 3) Electronic supplementary material: this should be contained in a separate file from the main text and the file name should contain the author's name and journal name, e.g. `authorname_procb_ESM_figures.pdf`

All supplementary materials accompanying an accepted article will be treated as in their final form. They will be published alongside the paper on the journal website and posted on the online figshare repository. Files on figshare will be made available approximately one week before the accompanying article so that the supplementary material can be attributed a unique DOI. Please see: <https://royalsociety.org/journals/authors/author-guidelines/>

- 4) Data-Sharing and data citation

It is a condition of publication that data supporting your paper are made available. Data should be made available either in the electronic supplementary material or through an appropriate repository. Details of how to access data should be included in your paper. Please see <https://royalsociety.org/journals/ethics-policies/data-sharing-mining/> for more details.

<http://datadryad.org/submit?journalID=RSPB&manu=RSPB-2019-1072> which will take you to your unique entry in the Dryad repository.

Once again, thank you for submitting your manuscript to Proceedings B and we look forward to receiving your final version. If you have any questions at all, please do not hesitate to get in touch.

Sincerely,

Proceedings B,
 mailto:proceedingsb@royalsociety.org
 =====

Reviewer Comments to Author:

Ruhr et al. provide a comprehensive insight into cardiomyocyte physiology of snapping turtles following normoxic or hypoxic embryonic incubation. I enjoyed reviewing this manuscript a second time and am pleased to see that my original concerns have been comprehensively addressed. The new statistical approach is much more appropriate and the new data on myofilament calcium sensitivity is more convincing. I wish to add no further comments.

Decision letter (RSPB-2019-1072.R1)

06-Jun-2019

Dear Dr Ruhr

I am pleased to inform you that your manuscript entitled "Developmental plasticity of cardiac anoxia-tolerance in juvenile common snapping turtles (*Chelydra serpentina*)" has been accepted for publication in Proceedings B.

Open Access

Paper charges

Sincerely,

Appendix A

Dear Prof. Braithwaite,

Many thanks for considering our manuscript, "Developmental plasticity of cardiac anoxia-tolerance in juvenile common snapping turtles, (*Chelydra serpentina*)" (RSPB-2019-0259). We would like to thank you for your helpful suggestions and the referees for their in-depth analysis and constructive comments on our manuscript. In the following document, we detail our responses in purple to each comment and refer to specific line numbers in the revised, track-changes document in red. We have reanalyzed the data with more appropriate GLMs (at your suggestion and that of Referee 1). Additionally, we have amended the text and figures, as suggested by the reviewers; the myofilament Ca²⁺-sensitivity data figures are now in the main text, whereas the graphs depicting morphometrics and the times to rise and half-decay have now been converted into tables. We believe the revised manuscript is much improved.

At your request, this new submission includes:

- 1) A clean copy of the revised manuscript.
- 2) A track-changes copy of the revised manuscript
- 3) A 'response to referees' document that details how we have responded to each of the referees' comments; adjustments to the manuscript are referenced with line numbers from the track-changes copy of the manuscript.
- 4) Line numbers in the main document.

We look forward to hearing from you in due course,

Best Wishes,
Ilan Ruhr

Editor

Both reviewers agree, as do I, that the MS is interesting, well written and provides valuable new information. They do however provide some suggestions to improve the MS, which should be considered in a revised version. In particular, I agree with Reviewer One that the statistical analyses need to be much more clearly described, and statistical support for the findings needs to be presented in the text. Currently a brief description of the statistical methodology and results are presented in the figure legends, rather than the methods and results. I think a one-way RMANOVA is appropriate for the data presented in Figs 3 & 4, but a more detailed description in the methods (i.e. indicating time is the repeat and hypoxia is the factor) would help. However, I don't understand the use of post hoc tests (my understanding is post hoc tests are not available for RMANOVA) and the between-group comparisons (this comparison can be provided by the factor, and as pair-wise comparisons needs a multiple comparison correction). Please provide statistical support for your findings in the results i.e. test statistics, model and error degrees for freedom, exact p value).

We thank the editor for their constructive comments and we agree that our statistical analysis could be improved. Therefore, we have reanalysed the data and provided a much more detailed description of our statistical methods (lines 170-188). Moreover, considering the concern about our initial statistical analyses, we consulted a statistician who scrutinized, and deemed appropriate, our new statistical approach. Following the statistician's guidance, our revised statistical methods section contains descriptions of our new analyses, using mixed-effects GLMs (see the response to comment 5 by Referee 1) and Sidak post-hoc tests for multiple comparison corrections. With regards to the latter, the statistician was unaware of any problems with the use of post-hoc tests for a RM-ANOVA.

Given the enormous amount of statistically significant data, we chose to include all test values (F-, t-, U-, and P-statistics) and degrees of freedom for the within- and between-group factors in the supplemental material (supplemental Tables S2 to S7), to maintain the readability of the manuscript. We believe our new statistical analyses are now much more thorough and descriptive.

Referee: 1

In this study, Ruhr and colleagues investigated the effect of hypoxic developmental programming (10 vs 21% O₂ exposure in ovo) on anoxia tolerance in cardiomyocytes of snapping turtles. The manuscript is eminently readable (the introduction is particularly beautifully written) and the authors must also be commended on figures that are pleasing to the eye. The methods are well described and I trust that the experiments were expertly performed.

The major conclusion (i.e. summed in the final paragraph, lines 276-278) is that the hypoxia programmed turtles have more tolerant hearts because they had greater myofilament calcium sensitivity and a better ability to suppress ROS. Unfortunately, I have both methodological and statistical concerns for the validity of these conclusions that I believe should be addressed.

We would like to wholeheartedly thank the reviewer for their kind words and comprehensive assessment of our MS, particularly for their praise about its readability, methodology, and aesthetics. We have carefully taken their comments into account and addressed them point-by-point below; when the manuscript has been altered from the original version, we refer to the line numbers in the track-changed document.

Major comments:

The major conclusion (i.e. summed in the final paragraph, lines 276-278) is that the hypoxia programmed turtles have more tolerant hearts because they had greater myofilament calcium sensitivity and a better ability to suppress ROS. Unfortunately, I have both methodological and statistical concerns for the validity of these conclusions that I believe should be addressed.

1. The authors claim to provide 'mechanistic insight' (line 68), but I believe the study is actually rather descriptive; although they are thorough in their descriptions, which include a variety of measurements, this doesn't equate to 'mechanistic'. This is even acknowledged by the authors on line 215-216 ('we did not investigate these mechanisms here'). I think this is a problem because some of the results, such as the huge (threefold) difference in calcium transient, are so staggering. Such findings would be more credible if the underlying mechanism for the differences could be shown (e.g. using a pharmacological approach to dissect calcium handling in the two groups and show how they are different).

We respectfully disagree with the reviewer that our results are not mechanistic. We have characterised the effects of simulated anoxia on cardiomyocyte shortening and investigated the mechanistic basis of that relationship by measuring changes in intracellular ion homeostasis and ROS production. As an example, we provide a mechanism for the reduction in cell shortening in the N21 group, by showing concurrent reductions in intracellular Ca²⁺ transients and pH; this is the first measurement of its kind (see next point). Our results also suggest the H10 group has enhanced myofilament calcium sensitivity, and we now provide further mechanistic evidence for this contention (see below). While we acknowledge that we didn't reveal all the mechanisms that underlie our results (including the differences in calcium transient dynamics), we feel that the present study has greatly improved our understanding and has set the stage for future work to delineate other mechanisms.

We would also like to stress that our study is the first to measure cell shortening simultaneously with $[Ca^{2+}]_i$, pH_i , and ROS in any ectothermic vertebrate. Furthermore, it is also the first study to investigate the cellular events that underlie ectothermic cardiomyocyte function during conditions that simulate anoxia and reoxygenation. We believe this is a valuable contribution to our field.

2. **Initially, I had trouble finding the data on myofilament Ca^{2+} sensitivity because it has been relegated to the ‘Supplementary Results’. Given that this is presented as an important finding, I am unsure why it is hidden like this if the authors have full confidence in it.**

We thank the referee for their suggestion; we have now moved the myofilament Ca^{2+} sensitivity data into the main manuscript (lines 248-258 and Fig. 4A, B) and we’ve performed additional experiments to support the calculations (lines 154-160 and Fig. 4C).

3. **At first sight, it appears reasonable that the cardiomyocytes of hypoxia programmed animals could have greater myofilament calcium sensitivity. Unfortunately, upon examining Figure 4, this quickly comes into doubt- the hypoxia tolerant animals had similar systolic Ca^{2+} but much higher diastolic Ca^{2+} - if their myofilaments were more sensitive to Ca^{2+} it could, in my mind, spell serious problems for cardiac relaxation (diastolic dysfunction), which is inconsistent with their apparently normal contractile phenotype.**

We understand the reviewers concerns regarding elevated diastolic calcium; this is known to be a trigger for arrhythmogenic events in mammalian cardiomyocytes. However, we believe this is very unlikely in turtle cardiomyocytes for the following reasons;

- (a) Turtle cardiomyocytes can tolerate extremely high levels of extracellular $[Ca^{2+}]_e$ during anoxic submergence (total blood plasma $[Ca^{2+}] = 46$ mM), without any conspicuous damage or arrhythmogenic activity. While intracellular Ca^{2+} was not recorded during these exposures, it seems very unlikely that they are able to defend against such high levels of external Ca^{2+} . Therefore, we do not believe a higher diastolic Ca^{2+} in the H10 cohort can be considered pathological. In this regard, it would be very interesting to understand the mechanisms that allow turtle cardiomyocytes to withstand calcium “overload”.
- (b) Furthermore, we measured the Ca^{2+} transient duration, during the course of our experiment; the times to rise and half-decay are the same for both N21 and H10 cells (now Table 2), which suggests the intracellular levels of diastolic calcium are not affecting contractile dynamics.
- (c) Lastly, following the reviewers recommendation for the new statistical analysis (i.e. mixed effects, repeated-measures GLMs), there are no significant differences in diastolic $[Ca^{2+}]_i$ between the two groups of cells ($P = 0.298$ at 5 min and 0.408 at 10 min).
4. **If the authors remain compelled that this supports their ‘mechanism’, there are a number of established methods for investigating myofilament Ca^{2+} sensitivity directly (skinned preparations, steady-state contractures). They resort to very tenuous reasoning in the Supplementary Results that quickly becomes circular (basically, as far as I can interpret it, the myofilaments must have been more sensitive because they have smaller calcium transients but similar force. This is not the equivalent to demonstrating increased calcium sensitivity).**

We thank the referee for their excellent suggestion. We have now included new data on myofilament Ca^{2+} sensitivity (Fig. 4C), where we subjected cardiomyocytes to different concentrations of extracellular Ca^{2+} , while measuring cell length, according to an experiment conducted by Wisløff et al., 2001 [1]. The new data show that an increase in extracellular Ca^{2+} leads to a greater degree of cell shortening in the H10 vs. N21 cells. This new data has been incorporated into the manuscript in both the methods and results sections (lines 154-160 & 248-258, and a new Fig 4C) and supports our initial position that the myofilaments of H10 cells are more sensitive to Ca^{2+} .

5. **The statistics are not thoroughly described and there are a few points where I would like to better understand how they have chosen their tests.**

Figure 3: There are two-factors: turtle programming (H or N) and acute anoxia exposure of the cells (i.e. time). The authors describe performing, separately, as far as I can see: “one-way, repeated-measures ANOVAs, followed by Holm-Sidak post-hoc tests, for within-group comparisons and individual Student’s t-tests or Mann-Whitney rank-sum tests for between-group comparisons.” Why did they choose to do this as opposed to analyzing all of the data together in a two-way ANOVA, or, to avoid the issue of incomplete data (some points are missing when the cells died), a mixed effects linear model? This would provide a much better insight into the data, complete with interactions.

We sincerely thank the reviewer for their very careful analysis of our statistical procedures, and their suggestions for improvement. We have rerun the stats using mixed-effects with the statistical program SPSS. In these new analyses, the between-group factor is developmental oxygen (N or H) and the within-group factors are time (5 → 40 min) and treatment (normoxia and anoxia). The new statistical analyses support the vast majority of our initial conclusions, with the following exceptions:

- (a) Within-group decreases in the H10 calcium transient are no longer significant, compared to the control period (Fig. 2B, supplemental Table S4).
- (b) Differences between N21 and H10 control levels of diastolic $[\text{Ca}^{2+}]$ (Fig. 3C, D, supplemental Table S5).
- (c) Differences between N21 and H10 pH_i at 15, 20, and 25 min ($P = 0.059, 0.069,$ and 0.052 at 15, 20, and 25 min, respectively) (Fig. 2D, supplemental Table S5).

6. **The biggest issue, I believe, with the present analysis is the use of separate t-tests for the between group comparisons- this seems inappropriate to me without a correction for multiple comparisons (e.g. Bonferroni correction).**

We agree with the reviewer’s assessment and have addressed this issue with the GLMs described above, in which we used Sidak tests to analyzed between-group, pairwise comparisons.

7. **Out of curiosity, I quickly analysed the raw data for ROS production with a linear mixed effects analysis followed by Bonferroni multiple comparisons (using GraphPad Prism) and believe there may be no significantly differences between the programming groups.**

We thank the reviewer for their rigorous appraisal of our data. After rerunning the stats with the mixed-effects, rm GLM, the significant differences in ROS production between the groups during anoxia are still present (Supplemental Table S5; $P = 0.043, 0.031, 0.028,$ and 0.016 at 15, 20, 25, and

30 min, respectively). The revised manuscript now reflects these new analyses in the Methods and Results sections (lines 170-185, 192-195, and 203-210, and Supplemental Tables S3, S4, and S5).

8. **As a lesser issue, when were the Student's t-tests or Mann-Whitney rank-sum tests used? Were the data tested for normality (this isn't stated)? If so, I think it is important to state which tests were used for which data.**

We thank the reviewer for pointing this out; we have now reported the normality tests in the manuscript (lines 169-171).

9. **Line 167-168: the authors say that ROS production was similar under (starting) control conditions, but all of the cells were normalised to 100% at the start of the trial, so of course they appeared similar during this initial period of normoxia? This naivety potentially highlights a more substantial concern that baseline ROS production could actually have been different at the start and we have no way of knowing, is there any way of truly indicating that the ROS production was similar at the start?**

We thank the reviewer for pointing out this inconsistency; we should have been more explicit in saying that we did not quantify *absolute* ROS levels, but rather looked at relative changes. This has been amended in the body to ensure clarity (lines 186-187 and 235-237). In order to calibrate ROS probes, known concentrations of the product must be produced in the cytosol of the cell. In this regard, DHE is very difficult to calibrate because it detects superoxide radical, rather than H₂O₂, which is a more stable product. Because of the instability of free radicals, it is incredibly difficult (and very expensive) to produce a standard curve of known concentrations of cardiomyocyte cytosolic superoxide. Due to these complications, the vast majority of studies on isolated cardiomyocytes express DHE in relative terms.

10. **To me it is unclear what condition the protocol, particularly with these saline solutions, is trying to simulate and how it is relevant for this species. The cardiomyocytes were exposed to a rapid and extreme decrease in HCO₃ from 35 to 15 mM which, combined with the increase in CO₂, causes a huge fall in pH (7.7 to 6.8).**

We agree with the reviewer that “true” anoxia involves several, simultaneous insults, including zero oxygen, metabolic & respiratory acidosis, hyperkalemia, hypercalcemia, and sympathetic stimulation. Rather than trying to mimic this complicated condition, which would be difficult to interpret, we were interested in challenging the cardiomyocytes with the three main insults; zero oxygen, respiratory acidosis and metabolic acidosis. The precise values were based on gas and plasma ion concentrations measured in *Trachemys scripta* after 4 hours of anoxia at 25°C. While we fully acknowledge that this is not a physiological approach, it allows us to compare tolerance limits between two phenotypes without too many confounding variables. We have now included this information in the manuscript, and we've referred to the condition as an “anoxic challenge” which we defined as zero oxygen combined with metabolic and respiratory acidosis (lines 96-109, 127, 234, 312, 378-379, 393, and 422).

Concerning the large decrease in [HCO₃⁻] from simulated normoxia to anoxia, these levels fall within the normal range of normoxic and anoxic turtle plasma from previous studies, including Reese et al., 2002. Moreover, although large differences in [HCO₃⁻] do affect cardiac function in fish, they are less pronounced in turtles (at least in western painted turtles) [2]. Indeed, in Jackson et al. (1991) cardiac

function is little different between normoxia-exposed painted turtles that were treated with distinct $[\text{HCO}_3^-]$ (40 vs 5 mM) [2]. This demonstrates that intracellular $[\text{HCO}_3^-]$ likely does not play a major role in cardiac function and it is anoxia that perturbs the heart.

- 11. The first, small, issue is that they only cite work performed on *Chrysemys*, which is quite distantly related from *Chelydra*. A relevant paper to consult would be: Reese, S.A., et al. 2015. *Physiol Biochem Zool* 75: 432–438. Which as far as I can see has not been cited. This, nonetheless, shows relatively similar changes to *Chrysemys*.**

We thank the reviewer for alerting us to this paper; we have now cited this publication in the manuscript (line 52).

- 12. However, more pressingly, most of these *in vivo* studies (included those already provided by the authors) are on the effects of chronic (typically days) of anoxia, how is this confluent with their acute 30 min exposure of the cardiomyocytes?**

We are not attempting to mimic the *in-vivo* condition; we are interested in revealing differences in the tolerance limits of the two cardiac phenotypes, which would be an indication of cardiac programming of anoxia tolerance. We are certainly interested in the cellular mechanisms underlying long-term anoxic survival, but it is not possible to maintain freshly isolated cardiomyocytes for extended periods of time (beyond 8 hours), as they start to experience physiological remodelling. Similarly, culturing ectothermic cardiomyocytes is not possible, because this leads to changes in the cellular phenotype. And lastly, untreated cardiomyocyte contractility deteriorates after approximately 60 mins; therefore, extended protocols can confound results.

Despite this limitation, we still strongly believe that our data is compelling, as the anoxia response of the normoxic-developed cardiomyocytes closely matches the whole-heart response of freshwater turtles (reviewed in [3-5] and most recently shown in [6]). This gives us confidence that we achieved anoxic conditions and the differences between the two phenotypes (H10 and N21) can be trusted.

- 13. Another paper that I believe has been overlooked by the authors, that is specific to their study species, is:**

Frische, S., Fago, A., Altimiras, J., 2000. Respiratory responses to short term hypoxia in the snapping turtle, *Chelydra serpentina*. *Comp. Biochem. Physiol., Part A Mol. Integr. Physiol.* 126, 223–231. doi:10.1016/S1095-6433(00)00201-4

Here it is shown that when this species is exposed to short term (aerial) hypoxia, ventilation increases meaning that plasma pH increases (opposite to the huge decrease in the protocol of the present paper). I think they need to better explain what sort of hypoxia they are interested in

We have now clarified and justified the components of the anoxic saline, detailing the type of anoxia in which we are interested (focussed on three main components: zero O_2 , CO_2 retention, and lactic-acid build-up; see point 14, below) that is distinct from the aerial hypoxia described in the Frische et al. paper.

- 14. As a distinct but related point, for the purposes of clearly understanding the response of the heart and providing comparisons to previous studies (some of which have used anoxia but no acidosis),**

I think it is important to emphasise in the main text that the present study explored combined anoxia and hypercapnic acidosis- the latter is hardly acknowledged. For example, on line 232 the authors report a 'profound intracellular acidosis', but this is not surprising in comparison to the even more profound extracellular acidosis.

We agree this distinction is important, and we have now included it in the manuscript (lines 101-107).

Minor comments:

- **For me, the terms N21 and H10 are something of a pleonasm. I don't think it is necessary to consistently reinforce that normoxia is 21% oxygen once it has been defined as such.**

We wish to be consistent with previous reports of developmental hypoxia in the American alligator and common snapping turtle, which use the abbreviations N21 and H10. In doing so, we believe it adds consistency across these studies, and adds a level of clarity for readers familiar with these reports.

- **Also, if, as described in the second paragraph of the introduction, eggs in the wild are often exposed to hypoxia, is 'normoxic' really 'normal' for these animals?**

This is an interesting point; we agree that there isn't a "normal" developmental oxygen level in this species because they can be subjected to a range of tensions in the wild. A brief explanation of why both hypoxia and normoxia can be considered "normal" has been included in the introduction (lines 44-45) and materials and methods (lines 85-87).

- **Line 133- It isn't patently clear what temperature the cells were exposed to in the closed cell bath (recording chamber)? I can see that they were loaded at room temperature, is the assumption then that they were also studied at that temperature?**

We thank the referee for pointing this out; we have clarified the temperature in the manuscript (lines 116, 147, and 159).

- **Line 134- how was this stimulation frequency chosen? It (i.e. 12 beats per minute) is approximately half the heart rate value reported in the two Wearing et al. (2016 and 2017) papers cited by the authors, although I would again encourage the authors to refer to Frische et al. (2000, CBP), where lower heart rates were observed.**

While the effects of anoxia on *in-vivo* cardiac function have not been measured in snapping turtles (to our knowledge), previous work has shown heart rate of *Trachemys scripta* decreases by 50% after 4 hours of anoxia at room temperature (Stecyk et al., AJP 2009). Therefore, we wanted to use a stimulation frequency that a turtle would likely encounter *in vivo* under normoxic and anoxic conditions. 12 BPM is within the normoxic range (10-25 BPM) and the estimated anoxic range for this species (5-12 BPM). While we acknowledge that our stimulation protocol does not recapitulate the effects of *in-vivo* anoxia, we chose to keep this parameter constant so we could interpret our findings in the absence of changes in frequency, which are known to independently affect contractility and calcium. We have now added this information to the supplementary methods section and referenced

the Frische paper, as well as other relevant papers (line 149 in main MS and line 59 in Supplemental Materials and Methods).

- **Line 191- was systolic Ca^{2+} not also still elevated?**

We agree and this has been clarified in the manuscript (line 245).

- **Line 269- a similar issue to above, you say intracellular Ca^{2+} returned to intermediate on pre-anoxic levels, but whilst this was true for the transient, it cannot be said for the systolic or diastolic concentrations per se.**

We agree; this has been clarified in the manuscript (lines 355).

Referee: 2

This manuscript by Ruhr et al reports an interesting study on how exposure to hypoxia during development protect cardiomyocytes from hypoxic stress later in life. The authors chose a well-suited species, the snapping turtle, a species highly tolerant of severe hypoxia, and appropriate combination of cutting edge techniques for monitoring cell morphology, shortening, intracellular Ca^{2+} and pH as well as ROS production of intact, isolated cardiomyocytes in a normoxia-anoxia-recovery protocol. The manuscript is very well written and the experiments have been carefully conducted. The conclusions that cardiomyocytes from hypoxic developing turtles contract more efficiently and produce less ROS are highly valuable and improve our understanding on the mechanisms controlling hypoxia tolerance of the vertebrate heart. I have only a few suggestions for the authors to consider, to improve this manuscript further.

We would like to sincerely thank the reviewer for their positive comments about our manuscript, particularly for its added value to the field of vertebrate hypoxia-tolerance. Their thorough evaluation of our MS has produced insightful suggestions for improvement, which we address point-by-point below; when the manuscript has been altered from the original version, we refer to the line numbers in the track-changed document.

- 1. Line 165: calcium transient kinetics. Make clear if these refer to traces A and B in Figure 4. In the same Figure 4, panels E and F, please eliminate the line connecting the data points since there are no intermediate conditions in between treatments.**

We thank the reviewer for their suggestions; we have made revisions to the figures and the manuscript, for clarity. The panels E and F, to which the reviewer refers, have now been consolidated into a new, solitary table (Table 1). We have also modified the data points to make bar graphs, thus, eliminating the connecting lines between the data.

- 2. Line 213 and 249: please add the reference to the figure in question, to help the reader.**

This has now been added to the manuscript (lines 268, 269, 276, 282, 289, 300, and 334).

- 3. Line 218: please explain how is Ca^{2+} sensitivity defined. Is it the Ca^{2+} concentration necessary to achieve a given contraction?**

We define myofilament calcium sensitivity as the relationship between the concentration of free calcium ions available for binding to Troponin C and the amount of force generated by the cardiomyocyte. This has now been added to the manuscript (lines 154-157).

- 4. In Figure 2, please add a color-code legend explaining that red bars refer to N21 and blue bars to H10**

We thank the referee for their suggestion; this figure has now been converted into Table 1.

- 5. In Figure 3, the much higher efficiency of contraction per unit of Ca^{2+} gradient of N10 turtles is a remarkable result in the context of energy-saving that however is not much discussed as it deserves.**

The reviewer makes an excellent point and we have briefly discussed the possibility of more efficient energy use by H10 cardiomyocytes (lines 298-308). We hope to study this more thoroughly in the future.

- 6. Also, given the role of Ca^{2+} as trigger of hypoxic cell death it would be interesting to know whether data of Figure 4 C or D could indicate (or perhaps not?) channel arrest, as defined by Hochachka?**

This is an interesting idea; channel arrest plays a major role in turtle neuron anoxia tolerance. However, Johnathan Stecyk's work on chronic anoxia suggests channel arrest does not occur in the heart, because this organ needs to be functional during anoxia to ensure efficient waste removal and nutrient supply. Furthermore, we imagine the brief period of anoxia in our study is unlikely to trigger downregulation of gene expression.

- 7. It seems that Ca^{2+} is leaking into cells over time (Fig. 4C)? I wonder whether the authors could expand more on these intriguing issues.**

The reviewer brings up an interesting point. Indeed, Ca^{2+} might be leaking into the cell, but this is more likely due to reverse NCX activity, whereby Na^+ is extruded from the cell for Ca^{2+} . This is a common signature of cardiomyocytes in anoxia. In mammals, the increase can be severe (into the millimolar range) leading to Ca^{2+} overload and death. In comparison, the progressive increases seen in our study are smaller and suggest that snapping turtles defend diastolic calcium levels. However, it is difficult to make comparisons between turtles and mammals, because of the different conditions by which the experiments are carried out (e.g. 37°C for mammals vs room temperature for turtles). For this reason, we chose not to expand upon these findings in the discussion.

1. **Wisløff, U., Loennechen, J. P., Falck, G., Beisvag, V., Currie, S., Smith, G. & Ellingsen, Ø.** 2001 Increased contractility and calcium sensitivity in cardiac myocytes isolated from endurance trained rats. *Cardiovasc Res* **50**, 495-508.
2. **Jackson, D. C., Arendt, E. A., Inman, K. C., Lawler, R. G., Panol, G. & Wasser, J. S.** 1991 ³¹P-NMR study of normoxic and anoxic perfused turtle heart during graded CO₂ and lactic acidosis. *Am J Physiol Regul Integr Comp Physiol* **260**, R1130-1136. (DOI:10.1152/ajpregu.1991.260.6.R1130).
3. **Jackson, D. C.** 1987 Cardiovascular function in turtles during anoxia and acidosis: *in-vivo* and *in-vitro* studies. *Am Zool* **27**, 49-58. (DOI:doi.org/10.1093/icb/27.1.49).
4. **Wasser, J. S.** 1996 Maintenance of cardiac function during anoxia in turtles: From cell to organism. *Comp Biochem Physiol B Biochem Mol Biol* **113**, 15-22.
5. **Farrell, A. P. & Stecyk, J. A. W.** 2007 The heart as a working model to explore themes and strategies for anoxic survival in ectothermic vertebrates. *Comp Biochem Phys A* **147**, 300-312. (DOI:10.1016/j.cbpa.2007.01.021).
6. **Bundgård, A., James, A. M., Joyce, W., Murphy, M. P. & Fago, A.** 2018 Suppression of reactive oxygen species generation in heart mitochondria from anoxic turtles: the role of complex I S-nitrosation. *J Exp Biol*, jeb. 174391. (DOI:10.1242/jeb.174391).